# The association between RDW-to-platelet ratio and in-hospital mortality in critically ill stroke patients: A retrospective cohort study based on the eICU database

Yu Chen◉*, Xiangrong Yang◉, Minmin Lan◉, Xinghua Qin◉, Dongmei Yi◉, Lu Chen◉

Department of Neurology, Guangxi Hydroelectric Hospital, Guilin, Guangxi, China

◉ These authors contributed equally to this work.
* yuchen123456112025@163.com

## Abstract

### Objective

The red cell distribution width-to-platelet ratio (RDW-to-platelet ratio, RPR) is a potential biomarker of inflammation and bone marrow function. This study aimed to investigate the association between RPR and in-hospital mortality in critically ill stroke patients in the eICU, and to evaluate mediating role of APACHE-IV score.

### Method

This retrospective cohort study utilized data from the 2014–2015 US multicenter eICU database. Of 200,859 initially included patients, 9,736 critically ill stroke patients were analyzed after excluding non-stroke cases and those with missing key variables. Multivariable logistic regression assessed the relationship between RPR and in-hospital mortality, adjusting for age, sex, comorbidities, and laboratory parameters, with multiple imputation for missing data. Subgroup analyses, mediation analysis, and restricted cubic spline modeling were performed.

### Results

Patients with higher RPR exhibited significantly elevated in-hospital mortality (17.7% vs. 11.6%, P < 0.001). After adjustment, each unit increase in RPR was associated with 4.6% higher odds of death (adjusted OR=1.046, 95% CI: 1.032–1.061, P < 0.001), with consistent findings across subgroups. Restricted cubic spline analysis indicated a linear relationship. Mediation analysis showed that APACHE-IV score mediated 20.15% of the total effect (P < 0.001).

**Data availability statement:** The datasets used and analyzed during the current study are derived from the eICU Collaborative Research Database, a publicly available and ethically approved critical care database. Researchers can access this database upon signing a data use agreement and completing the required training. The specific data extraction of this study were uploaded.

**Funding:** The author(s) received no specific funding for this work.

**Competing interests:** The authors have declared that no competing interests exist.

## Conclusions

Elevated RPR is independently associated in-hospital mortality in critically ill stroke patients, partly mediated by disease severity. As a simple and accessible marker, RPR shows promise for clinical prognostic risk stratification.

## Introduction

In-hospital mortality among stroke patients in the intensive care unit (ICU) reaches 25%–35%, escalating to 40%–50% for hemorrhagic stroke, and exceeding 50% when complicated by heart failure or acute myocardial infarction (AMI) [1]. In the United States, ICU-admitted stroke patients comprise 5%–10% of all hospitalized stroke cases, with over 70% comorbid with hypertension, representing a predominant clinical subgroup [2]. These patients impose a substantial disease burden, with average hospital stays of 7–14 days and high resource utilization, underscoring the need for reliable early prognostic markers to inform clinical decisions.

Red cell distribution width (RDW), a routine complete blood count parameter reflecting erythrocyte volume heterogeneity, has been linked to systemic inflammation, oxidative stress, and endothelial dysfunction [3,4]. Mechanistically, elevated RDW indicates dysregulated erythropoiesis in bone marrow under the influence of inflammatory cytokines such as IL-6 and TNF-α, resulting in impaired oxygen-carrying capacity, microcirculatory dysfunction, and tissue hypoxia [5]. Furthermore, RDW correlates with hypoalbuminemia, malnutrition, and immunosuppression [6]. The RDW-to-platelet ratio (RPR), integrating RDW with platelet count, captures dual pathophysiological processes of inflammation and coagulopathy, providing a comprehensive reflection of stress in critically ill patients [7].

This study, utilizing the eICU database, conducts a retrospective cohort analysis to examine the association between RPR and in-hospital mortality in ICU stroke patients, adjusting for confounders including age, sex, APACHE-IV score, baseline comorbidities (e.g., diabetes, malignancy), laboratory parameters (blood urea nitrogen, albumin, hemoglobin, creatinine), and therapeutic interventions (anticoagulation/antiplatelet therapy). We aim to establish RPR as an independent association, supporting its application in clinical risk stratification.

## Methods

### Study design and data source

This multicenter retrospective cohort study was conducted using the eICU Collaborative Research Database (eICU-CRD) version 2.0, a publicly available multi-center critical care database comprising de-identified electronic health records from over 200,000 admissions to 208 intensive care units (ICUs) across the continental United States, spanning 2014, to 2015. The study population consisted of adult critically ill patients with a confirmed diagnosis of stroke (ischemic or hemorrhagic) during their ICU stay. Inclusion criteria were: (1) age ≥ 18 years; (2) stroke diagnosis confirmed during ICU admission; and (3) availability of complete laboratory data within the first

24 hours of admission. Exclusion criteria included: (1) admissions not primarily related to stroke; (2) missing data on red cell distribution width (RDW)-to-platelet ratio; and (3) missing in-hospital mortality data. From the initial cohort of 200,859 patients, 9,736 met the eligibility criteria and were included in the analyses (Fig 1). Clinical variables were extracted automatically from standardized electronic health records and verified independently by two trained researchers to ensure data integrity. Stroke diagnoses were ascertained using International Classification of Diseases, Ninth Revision (ICD-9) codes supplemented by neuroimaging reports.

## Exposure, outcome, and covariates

The primary exposure variable was the RDW-to-platelet ratio (RPR), defined as the initial Red Cell Distribution Width (RDW) value (expressed as a percentage, %) obtained within 24 hours of ICU admission divided by the platelet count (expressed as $\times 10^9$/L). For example, if a patient's RDW was 15.5% and platelet count was $250 \times 10^9$/L, the RPR would be calculated as $15.5 \div 250 = 0.062$. Data were recorded as continuous numerical values. The primary outcome was in-hospital mortality from any cause during the index hospitalization, determined from electronic hospital death registries and structured discharge status fields (coded as alive versus deceased). Adjudication was based on objective, structured clinical data without blinding, thereby reducing potential subjective bias.

   Covariate selection was guided by the following systematic principles: (1) Demographic variables (age, sex, race/ethnicity) were included as well-established independent prognostic factors for stroke patients; (2) Disease severity indicators (Acute Physiology and Chronic Health Evaluation IV [APACHE-IV] score) were selected for their recognized association in critically ill patients for assessing acute physiological status; (3) Comorbidities (congestive heart failure, hypertension, diabetes mellitus, malignancy, sepsis) were incorporated based on extensive evidence linking these chronic conditions to

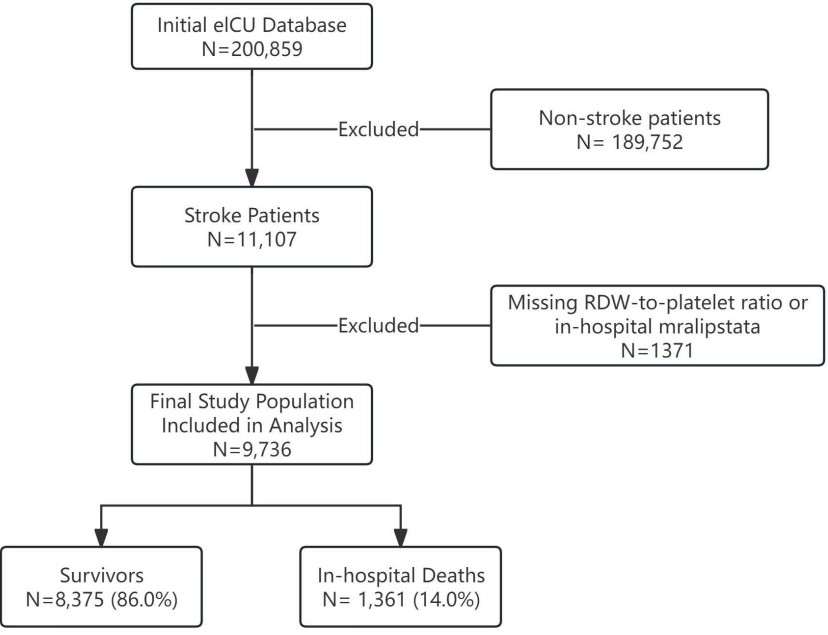

**Fig 1. Study population selection flowchart.** The figure illustrates the sequential patient selection process from the initial eICU database (N = 200,859) through application of inclusion and exclusion criteria to the final analytic cohort (N = 9,736). The flowchart shows: (1) Initial screening of all eICU patients, with exclusion of non-stroke admissions (n = 189,752); (2) Identification of stroke patients (n = 11,107); (3) Exclusion of those with missing key variables including RDW-to-platelet ratio or in-hospital mortality data (n = 1,371); and (4) Final classification of patients into outcome categories (survivors: n = 8,375; in-hospital deaths: n = 1,361). Abbreviation: eICU, eICU Collaborative Research Database; RDW, red cell distribution width.

adverse outcomes in stroke populations; (4) Therapeutic interventions (mechanical ventilation, anticoagulation, antiplatelet therapy) were included to capture clinical management intensity and underlying physiological derangement; (5) Laboratory parameters (albumin, blood urea nitrogen, hemoglobin, creatinine) were selected as they reflect nutritional status, renal function, anemia, and inflammation—each independently associated with mortality in critically ill patients across multiple studies in the MIMIC and eICU databases. All covariates were simultaneously included in the model to comprehensively adjust for potential confounding while minimizing the risk of over-adjustment, as verified by variance inflation factors (VIF < 5 for all variables).

The selected covariates were chosen based on three principal considerations: First, substantial evidence from clinical literature—we systematically reviewed prognostic studies in critically ill stroke patients and confirmed that all included variables have been established as independent association of mortality in multiple investigations. Second, biological and clinical relevance—each variable either directly reflects disease biology (e.g., RDW as a marker of inflammation and bone marrow dysfunction) or represents an important clinical characteristic (e.g., APACHE-IV score quantifying illness severity). Third, data completeness and statistical feasibility with a sample size of 9,736 patients, the number of covariates was balanced to adequately control for confounding while maintaining statistical power. For variables with ≥5% missing values, we employed the missforest algorithm for multiple imputation, which outperforms traditional methods in handling complex missing data patterns and was validated through sensitivity analyses demonstrating robustness of results.

### Ethical considerations

The study protocol was approved by the Institutional Review Board (IRB approval No. IRB-2023-EICU-001). Informed consent was waived due to the use of de-identified retrospective data. Data access complied with PhysioNet credentialing requirements, with all records anonymized prior to analysis. Secure, encrypted platforms restricted access to authorized investigators only. The investigation adhered to the principles of the Declaration of Helsinki and eICU-CRD data-sharing policies.

### Statistical analysis

We described baseline characteristics using means with standard deviations for continuous variables and frequencies with percentages for categorical variables, stratified by tertiles of the RDW-to-platelet ratio. Continuous variables across tertiles were compared using one-way analysis of variance (ANOVA), and categorical variables using the $\chi^2$ test; trends were assessed with the Cochran-Armitage test for categorical variables and Jonckheere-Terpstra test for continuous variables (all P values two-sided).

In-hospital mortality served as the primary outcome. We first performed univariate logistic regression to assess the crude association between RDW-to-platelet ratio (modeled as a continuous variable) and in-hospital mortality, reporting odds ratios (ORs) with 95% confidence intervals (CIs). Multivariable logistic regression models were constructed in a stepwise manner, adjusting for age, sex, comorbidities (e.g., congestive heart failure, malignancy, diabetes), laboratory values (e.g., albumin, hemoglobin, blood urea nitrogen, creatinine), and clinical variables (e.g., Glasgow Coma Scale score, mechanical ventilation, sepsis, APACHE-IV score). Adjusted ORs (aORs) with 95% CIs were reported per unit increase in RDW-to-platelet ratio.

To address missing data (primarily albumin [29.21%] and APACHE-IV score [14.50%]; < 5% for others) (S1 Table), we used multiple imputation by chained equations with five iterations via the missForest algorithm, incorporating all covariates under a random forest framework. Pooled estimates were obtained using Rubin's rules. Variance inflation factors (VIFs) were calculated to assess multicollinearity (all VIFs < 5).

Nonlinearity between RDW-to-platelet ratio and in-hospital mortality was examined using restricted cubic splines with three knots at the 10th, 50th, and 90th percentiles, fitted within a multivariable logistic regression model; the reference level was set at the median value.

Mediation analysis was conducted using the mediation R package, following the general approach to causal mediation analysis. The RDW-to-platelet ratio was treated as the exposure, APACHE-IV score as the mediator, and in-hospital mortality as the outcome, with adjustment for age, ethnicity, sex, congestive heart failure, diabetes, sepsis, malignant tumor burden, albumin, creatinine, and mechanical ventilation status. Nonparametric bootstrap resampling (1000 iterations) with the percentile method was used to estimate confidence intervals for the total effect, mediation (indirect) effect, direct effect, and proportion mediated.

Subgroup analyses stratified by sex, age groups (<65 vs. ≥65 years), race, mechanical ventilation status, comorbidities, and APACHE-IV tertiles were performed by including interaction terms in the multivariable logistic regression model; effect modification was tested using the likelihood ratio test (P < 0.05 indicating significance). All analyses were performed using R version 4.3.1 (packages: mice, missForest, rms, mediation).

To handle missing values in our dataset, we employed the missForest multiple imputation algorithm, which is based on random forests. This algorithm iteratively imputes missing data by building a random forest model for each variable with missing values, using all other variables as predictors to estimate the missing entries. We set the maximum number of iterations to 5 (maxiter = 5) to ensure convergence. For each random forest model constructed, we used 100 decision trees (ntree = 100). The number of variables randomly sampled as candidates at each split was set to the square root of the total number of predictor variables (mtry = sqrt(p), where p is the total number of predictors). To further enhance the stability of the imputation process, variables were imputed in decreasing order of the number of missing values (decreasing = TRUE). The convergence of the imputation process was confirmed by monitoring the changes in Out-of-Bag (OOB) error, with stabilization indicating model convergence. Detailed parameter settings and convergence diagnostics of the missForest algorithm are provided in Supplemental S2 Table. The quality of the imputed data was assessed by comparing the distributions of key variables between the original and imputed datasets. As shown in Supplemental S3 Table, no significant differences were observed in the distributions of key variables before and after imputation, confirming the rationality and reliability of the imputation method.

## Results

This study included 200,859 patients from the eICU database, yielding 9,736 adult stroke patients with critical illness after exclusions (study flowchart, Fig 1). Baseline characteristics stratified by RDW-to-platelet ratio tertiles are summarized in Table 1. The cohort had a mean age of 67.06 ± 14.86 years, with 52.0% male and 14.0% in-hospital mortality. Across tertiles (low to high), significant trends were observed toward older age (63.87 vs. 67.29 vs. 70.01 years), higher APACHE-IV scores (51.97 vs. 52.82 vs. 59.48), increased mechanical ventilation use (21.9% vs. 22.2% vs. 26.7%), greater prevalence of congestive heart failure and malignancy, lower albumin and hemoglobin levels, and elevated blood urea nitrogen and creatinine (all P < 0.05). In-hospital mortality rose significantly from 11.6% in the low-ratio group to 17.7% in the high-ratio group (P < 0.001).

Univariate logistic regression revealed that each unit increase in RDW-to-platelet ratio was associated with a 7% higher risk of in-hospital mortality (OR=1.07, 95% CI: 1.056–1.080, P < 0.001), with an effect size exceeding that of age (OR=1.02) and diabetes (OR=1.27). Other significant variables included lower GCS scores, mechanical ventilation, sepsis, anemia, and renal impairment (all P < 0.05)) (Table 2). In multivariable logistic regression adjusted for age, sex, comorbidities, laboratory values, and other clinical variables, the RDW-to-platelet ratio remained independently associated with in-hospital mortality (aOR=1.046 per unit increase, 95% CI: 1.032–1.061, P < 0.001) (Table 3). Sensitivity analyses using multiple imputation (five iterations) yielded consistent results (pooled estimates in S4 Table), confirming robustness. Variance inflation factors were all < 5, indicating no substantial multicollinearity. To assess nonlinearity, a smoothed curve was fitted for the association between RDW-to-platelet ratio and in-hospital mortality (Fig 2). The relationship showed an approximately linear trend. Mediation analysis demonstrated that the APACHE-IV score partially mediated the association between RDW-to-platelet ratio and in-hospital mortality (Table 4, Fig 3). Comparing high- versus low-ratio groups (8.25

**Table 1. Baseline characteristics of study participants by RDW-to-platelet ratio.**

| Characteristic | Overall | Low | Medium | High | P Value |
|---|---|---|---|---|---|
| Sample size | | | | | |
| n | 9,736 | 3,246 [1.16, 5.67] | 3,245 [5.67, 7.56] | 3,245 [7.56, 55.00] | |
| In-hospital mortality | | | | | <0.001 |
| Alive | 8,375 (86.0%) | 2,870 (88.4%) | 2,834 (87.3%) | 2,671 (82.3%) | |
| Dead | 1,361 (14.0%) | 376 (11.6%) | 411 (12.7%) | 574 (17.7%) | |
| Total GCS score | 12.54±3.55 | 12.59±3.57 | 12.65±3.47 | 12.39±3.60 | 0.009 |
| Age, years | 67.06±14.86 | 63.87±15.52 | 67.29±14.65 | 70.01±13.72 | <0.001 |
| Gender | | | | | <0.001 |
| Male | 5,058 (52.0%) | 1,396 (43.0%) | 1,722 (53.1%) | 1,940 (59.8%) | |
| Female | 4,677 (48.0%) | 1,850 (57.0%) | 1,522 (46.9%) | 1,305 (40.2%) | |
| APACHE-IV score | 54.74±25.17 | 51.97±24.71 | 52.82±24.50 | 59.48±25.64 | <0.001 |
| Ethnicity | | | | | 0.005 |
| African American | 1,188 (12.2%) | 372 (11.5%) | 389 (12.0%) | 427 (13.2%) | |
| Asian | 208 (2.1%) | 71 (2.2%) | 66 (2.0%) | 71 (2.2%) | |
| Caucasian | 7,326 (75.2%) | 2,473 (76.2%) | 2,406 (74.1%) | 2,447 (75.4%) | |
| Hispanic | 397 (4.1%) | 110 (3.4%) | 166 (5.1%) | 121 (3.7%) | |
| Native American | 45 (0.5%) | 17 (0.5%) | 14 (0.4%) | 14 (0.4%) | |
| Other/Unknown | 525 (5.4%) | 190 (5.9%) | 187 (5.8%) | 148 (4.6%) | |
| Mechanical ventilation | | | | | <0.001 |
| No | 7,436 (76.4%) | 2,535 (78.1%) | 2,523 (77.8%) | 2,378 (73.3%) | |
| Yes | 2,300 (23.6%) | 711 (21.9%) | 722 (22.2%) | 867 (26.7%) | |
| Congestive heart failure | | | | | <0.001 |
| No | 9,244 (94.9%) | 3,137 (96.6%) | 3,085 (95.1%) | 3,022 (93.1%) | |
| Yes | 492 (5.1%) | 109 (3.4%) | 160 (4.9%) | 223 (6.9%) | |
| Hypertension | | | | | 0.048 |
| No | 6,646 (68.3%) | 2,246 (69.2%) | 2,162 (66.6%) | 2,238 (69.0%) | |
| Yes | 3,090 (31.7%) | 1,000 (30.8%) | 1,083 (33.4%) | 1,007 (31.0%) | |
| Diabetes mellitus | | | | | 0.066 |
| No | 8,434 (86.6%) | 2,837 (87.4%) | 2,822 (87.0%) | 2,775 (85.5%) | |
| Yes | 1,302 (13.4%) | 409 (12.6%) | 423 (13.0%) | 470 (14.5%) | |
| Malignant tumor | | | | | <0.001 |
| No | 9,655 (99.2%) | 3,235 (99.7%) | 3,227 (99.4%) | 3,193 (98.4%) | |
| Yes | 81 (0.8%) | 11 (0.3%) | 18 (0.6%) | 52 (1.6%) | |
| Sepsis | | | | | <0.001 |
| No | 9,143 (93.9%) | 3,077 (94.8%) | 3,097 (95.4%) | 2,969 (91.5%) | |
| Yes | 593 (6.1%) | 169 (5.2%) | 148 (4.6%) | 276 (8.5%) | |
| Blood urea nitrogen, mg/dL | 18.00 (13.00, 25.00) | 16.00 (12.00, 23.00) | 17.00 (13.00, 24.00) | 20.00 (14.00, 29.00) | <0.001 |
| Albumin, g/dL | 3.41±0.67 | 3.49±0.67 | 3.49±0.63 | 3.26±0.67 | <0.001 |
| Hemoglobin, g/dL | 12.77±2.30 | 12.98±2.25 | 13.05±2.16 | 12.28±2.42 | <0.001 |
| Creatinine, mg/dL | 0.95 (0.74, 1.28) | 0.88 (0.70, 1.14) | 0.94 (0.75, 1.22) | 1.05 (0.80, 1.47) | <0.001 |
| Anticoagulant therapy | | | | | 0.112 |
| No | 9,548 (98.1%) | 3,191 (98.3%) | 3,169 (97.7%) | 3,188 (98.2%) | |
| Yes | 188 (1.9%) | 55 (1.7%) | 76 (2.3%) | 57 (1.8%) | |

*(Continued)*

**Table 1.** (Continued)

| Characteristic | Overall | Low | Medium | High | P Value |
|---|---|---|---|---|---|
| Antiplatelet therapy | | | | | 0.897 |
| No | 9,137 (93.8%) | 3,046 (93.8%) | 3,050 (94.0%) | 3,041 (93.7%) | |
| Yes | 599 (6.2%) | 200 (6.2%) | 195 (6.0%) | 204 (6.3%) | |

Values are n (%) for categorical variables and mean ± SD or median (IQR) for continuous variables.

Abbreviations: SD, standard deviation; IQR, interquartile range; GCS, Glasgow Coma Scale; APACHE, Acute Physiology and Chronic Health Evaluation.

P values were derived from χ² tests for categorical variables, one-way ANOVA for normally distributed continuous variables, and Kruskal-Wallis H tests for non-normally distributed continuous variables.

vs. 5.23), the total effect was an absolute risk difference of 2.35% (95% CI: 1.54%–3.21%), comprising a direct effect of 1.88% (95% CI: 1.05%–2.72%) and an indirect effect via APACHE-IV of 0.47% (95% CI: 0.25%–0.71%), accounting for 20.15% of the total effect (95% CI: 10.43%–34.89%; all paths $P < 0.001$). This indicates that disease severity contributes substantially to the underlying biological pathway.

Subgroup analyses confirmed the main effect's consistency across most strata (Table 5; all $P < 0.05$ for RDW-to-platelet ratio–mortality interactions), including sex, age groups, race, mechanical ventilation status, and comorbidities. Significant heterogeneity was noted by APACHE-IV tertiles ($P_{interaction} < 0.001$), with stronger associations in lower-risk patients (OR = 1.118, 95% CI: 1.074–1.165) than in higher-risk patients (effect attenuated but significant). Modest effect modification was also evident by sex ($P_{interaction} = 0.038$) and mechanical ventilation ($P_{interaction} = 0.003$), with slightly stronger associations in females and non-ventilated patients. Missing data analysis revealed primary deficits in albumin (29.21%) and APACHE-IV scores (14.50%), with other covariates <5% missing (S1 Table). Multiple imputation via the missForest algorithm produced imputed datasets comparable to originals across key variables (all $P > 0.05$; S2 Table), indicating no systematic bias and high data integrity.

## Discussion

This retrospective cohort analysis, utilizing the eICU database, examined 9,736 critically ill stroke patients to explore the association between the red cell distribution width-to-platelet ratio (RDW-to-platelet ratio, RPR) and in-hospital mortality risk. Results revealed a significant positive correlation; in multivariable-adjusted models, each 1-unit increase in RPR conferred a 4.6% higher mortality risk (OR = 1.046, 95% CI: 1.032–1.061, $P < 0.001$). Stratified analyses affirmed consistency across subgroups, with amplified effects among females, those with elevated APACHE-IV scores, and patients without diabetes. Mediation analysis further disclosed that the APACHE-IV score partially mediated the RPR-mortality relationship, accounting for approximately 20.15% of the total effect, implying a pivotal intermediary role for disease severity in this biological pathway.

As an index combining red cell distribution width (RDW) and platelet count (PLT), RPR may participate in the pathophysiology of stroke through multiple pathways. Elevated RDW is often associated with systemic inflammatory responses, oxidative stress, and endothelial dysfunction, all of which exacerbate brain injury and neuronal apoptosis after stroke. Concurrently, platelet count directly reflects coagulation function; stroke patients often present with hypercoagulable states or platelet dysfunction, and changes in RPR may indicate an imbalance in the coagulation-fibrinolysis system, thereby affecting thrombus formation, reperfusion injury, or bleeding risk. Furthermore, RPR, as an indicator of red blood cell volume heterogeneity, may be related to red blood cell deformability and insufficient microcirculatory perfusion, particularly in ischemic brain tissue, which further aggravates tissue hypoxia and damage.

Our findings converge with multiple investigations leveraging MIMIC and eICU databases, supports the association between RPR and in-hospital mortality in stroke-critical care. Chen et al. reported that RPR independently forecasted

**Table 2. Univariate analysis for in-hospital mortality.**

| Variable | OR | 95% CI | P Value |
|---|---|---|---|
| RDW-to-platelet ratio | 1.07 | (1.056, 1.080) | <0.001 |
| Total GCS score | 0.77 | (0.762, 0.785) | <0.001 |
| Age, years | 1.02 | (1.012, 1.020) | <0.001 |
| Gender | | | |
| Female vs Male | 1.02 | (0.911, 1.145) | 0.720 |
| APACHE-IV score | 1.05 | (1.044, 1.049) | <0.001 |
| Ethnicity | | | |
| Asian vs African American | 1.17 | (0.769, 1.772) | 0.468 |
| Caucasian vs African American | 1.08 | (0.897, 1.289) | 0.431 |
| Hispanic vs African American | 1.23 | (0.896, 1.697) | 0.198 |
| Native American vs African American | 0.48 | (0.146, 1.554) | 0.219 |
| Other/Unknown vs African American | 1.25 | (0.937, 1.671) | 0.128 |
| Mechanical ventilation | | | |
| Yes vs No | 7.94 | (7.022, 8.988) | <0.001 |
| Congestive heart failure | | | |
| Yes vs No | 1.77 | (1.418, 2.212) | <0.001 |
| Hypertension | | | |
| Yes vs No | 0.90 | (0.792, 1.017) | 0.091 |
| Diabetes mellitus | | | |
| Yes vs No | 1.27 | (1.080, 1.484) | 0.004 |
| Malignant tumor | | | |
| Yes vs No | 2.03 | (1.223, 3.379) | 0.006 |
| Sepsis | | | |
| Yes vs No | 2.66 | (2.202, 3.210) | <0.001 |
| Blood urea nitrogen, mg/dL | 1.01 | (1.012, 1.018) | <0.001 |
| Albumin, g/dL | 0.57 | (0.515, 0.623) | <0.001 |
| Hemoglobin, g/dL | 0.93 | (0.910, 0.956) | <0.001 |
| Creatinine, mg/dL | 1.11 | (1.066, 1.147) | <0.001 |
| Anticoagulant therapy | | | |
| Yes vs No | 0.90 | (0.583, 1.385) | 0.628 |
| Antiplatelet therapy | | | |
| Yes vs No | 0.90 | (0.705, 1.154) | 0.413 |

Abbreviations: OR, odds ratio; CI, confidence interval; GCS, Glasgow Coma Scale; APACHE, Acute Physiology and Chronic Health Evaluation.

Statistical significance was set at P < 0.05 (two-sided).

in-hospital mortality in patients with acute exacerbations of chronic obstructive pulmonary disease (HR = 1.47, 95% CI: 1.29–1.68), persisting after adjustment for APACHE scores and inflammatory markers. Huang et al. [2] observed in hypertensive stroke patients that each 1% RDW elevation heightened 28-day mortality by 18% (OR = 1.18, 95% CI: 1.12–1.25), bolstering RDW's prognostic utility as a systemic inflammation surrogate. Although these did not directly utilize RPR, mechanistic alignment is evident: RDW signifies chronic inflammation and erythrocytic anisocytosis, while platelet count denotes coagulopathy; their ratio holistically gauges systemic stress in critical illness. Moreover, Guo et al. [1] demonstrated that acute myocardial infarction augmented RDW's mortality predictive capacity in stroke (interaction P = 0.03), underscoring comorbidity modulation. Although we eschewed stratification by acute myocardial infarction, confounder adjustments for malignancy and infection enhanced robustness. Contemporary evidence reinforces RPR's role in stroke

**Table 3. Multivariable regression analysis of RDW-to-platelet ratio associated with in-hospital mortality.**

| Characteristic | Unadjusted OR (95% CI) | P Value | Adjusted OR (95% CI)† | P Value† |
|---|---|---|---|---|
| RDW-to-platelet ratio | 1.068 (1.056, 1.080) | <0.001 | 1.046 (1.032, 1.061) | <0.001 |

Abbreviations: OR, odds ratio; CI, confidence interval.

Adjusted for age, sex, comorbidities, laboratory parameters, and other clinically relevant variables.

Variance inflation factor (VIF) for all variables was < 5, indicating absence of multicollinearity.

† Values derived from multiple imputation for missing data.

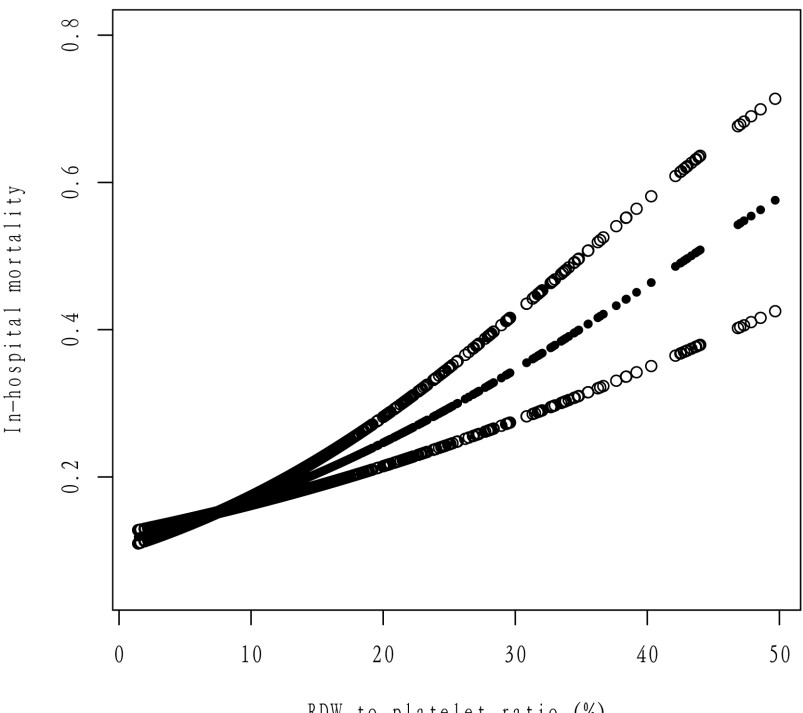

**Fig 2. RDW to platelet ratio (%).**

**Table 4. Mediation analysis of APACHE-IV score in the association between RDW-to-platelet ratio and in-hospital mortality.**

| Effect | Estimate | 95% CI | P Value |
|---|---|---|---|
| Total Effect | 2.35% | (1.54%, 3.21%) | <0.001 |
| Direct Effect | 1.88% | (1.05%, 2.72%) | <0.001 |
| Indirect Effect (Mediated) | 0.47% | (0.25%, 0.71%) | <0.001 |
| Proportion Mediated | 20.15% | (10.43%, 34.89%) | <0.001 |

Effects represent absolute risk differences comparing high vs low RDW-to-platelet ratio (8.25 vs 5.23). Analysis adjusted for age, ethnicity, gender, congestive heart failure, diabetes mellitus, sepsis, malignant tumor, albumin, creatinine, and mechanical ventilation. Confidence intervals calculated using nonparametric bootstrap with percentile method (1,000 resamples).

Abbreviations: CI, confidence interval; RDW, red cell distribution width; APACHE, Acute Physiology and Chronic Health Evaluation.

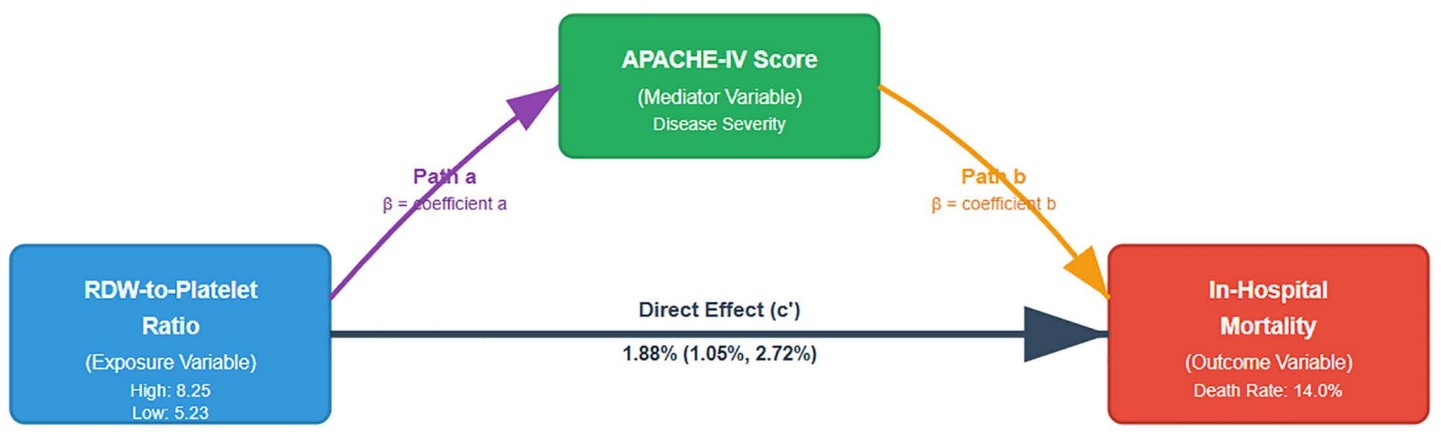

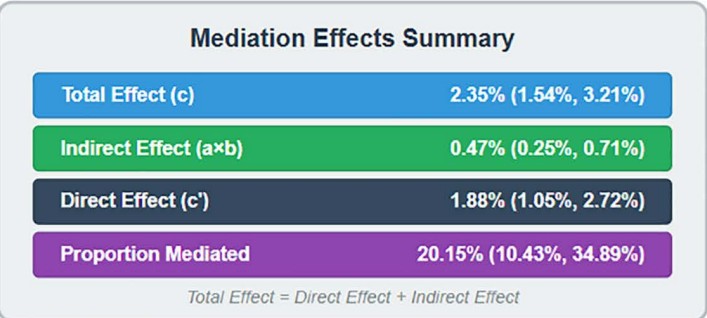

**Fig 3. Mediation analysis pathway diagram.**

prognostication: Liang et al. [8] linked elevated admission RPR to heightened 30-day mortality in spontaneous intracerebral hemorrhage (highest tertile HR = 1.37, 95% CI: 1.15–1.64), albeit non-significant in low comorbidity index (CCI 3 mg/L) subgroups (OR = 3.6, 95% CI: 1.6–8.3), with no significant difference in low-CRP groups (P = 0.6), underscoring RPR's context-dependent prognostic value. These convergences substantiate RPR's validity associated with stroke-critical care, while highlighting its amplified utility in high-risk subgroups, aligning with our high-APACHE-IV effects.

Methodologically, this study harbors several strengths. Foremost, sourcing from the multicenter eICU database encapsulates real-world ICU dynamics, augmenting external validity. Second, multiple imputation via the missForest algorithm mitigated missing data, with sensitivity analyses evincing negligible baseline disparities pre- and post-imputation (all P > 0.05) [S2 Table], thereby curtailing selection bias. Third, restricted cubic splines and smoothed curve fitting corroborated the linearity of the RPR-mortality nexus, forestalling overinterpretation inherent to nonlinear paradigms. Finally,

**Table 5. Stratified analysis of RDW-to-platelet ratio and in-hospital mortality association.**

| Stratification Variable | Stratum | n | OR (95% CI) | P Value | P for interaction |
|---|---|---|---|---|---|
| Total GCS score | | | | | 0.015 |
| | Low | 4,826 | 1.062 (1.047, 1.077) | <0.001 | |
| | High | 4,656 | 1.068 (1.044, 1.092) | <0.001 | |
| Age, years | | | | | 0.388 |
| | Low | 3,305 | 1.053 (1.034, 1.072) | <0.001 | |
| | Medium | 3,261 | 1.084 (1.063, 1.105) | <0.001 | |
| | High | 3,170 | 1.067 (1.043, 1.092) | <0.001 | |
| Gender | | | | | 0.038 |
| | Male | 5,058 | 1.057 (1.041, 1.073) | <0.001 | |
| | Female | 4,677 | 1.083 (1.064, 1.103) | <0.001 | |
| APACHE-IV score | | | | | <0.001 |
| | Low | 2,776 | 1.118 (1.074, 1.165) | <0.001 | |
| | Medium | 2,862 | 1.047 (1.014, 1.080) | 0.004 | |
| | High | 2,686 | 1.032 (1.016, 1.048) | <0.001 | |
| Ethnicity | | | | | 0.171 |
| | African American | 1,188 | 1.085 (1.053, 1.117) | <0.001 | |
| | Asian | 208 | 1.019 (0.918, 1.131) | 0.719 | |
| | Caucasian | 7,326 | 1.060 (1.046, 1.074) | <0.001 | |
| | Hispanic | 397 | 1.100 (1.043, 1.160) | <0.001 | |
| | Native American | 45 | 0.934 (0.597, 1.461) | 0.765 | |
| | Other/Unknown | 525 | 1.116 (1.058, 1.178) | <0.001 | |
| Mechanical ventilation | | | | | 0.003 |
| | No | 7,436 | 1.077 (1.058, 1.095) | <0.001 | |
| | Yes | 2,300 | 1.038 (1.022, 1.055) | <0.001 | |
| Congestive heart failure | | | | | 0.866 |
| | No | 9,244 | 1.067 (1.055, 1.079) | <0.001 | |
| | Yes | 492 | 1.062 (1.011, 1.117) | 0.018 | |
| Hypertension | | | | | 0.172 |
| | No | 6,646 | 1.072 (1.058, 1.087) | <0.001 | |
| | Yes | 3,090 | 1.053 (1.030, 1.077) | <0.001 | |
| Diabetes mellitus | | | | | 0.064 |
| | No | 8,434 | 1.072 (1.059, 1.086) | <0.001 | |
| | Yes | 1,302 | 1.040 (1.010, 1.072) | 0.009 | |
| Malignant tumor | | | | | 0.872 |
| | No | 9,655 | 1.067 (1.055, 1.079) | <0.001 | |
| | Yes | 81 | 1.061 (0.992, 1.135) | 0.086 | |
| Sepsis | | | | | 0.006 |
| | No | 9,143 | 1.070 (1.057, 1.084) | <0.001 | |
| | Yes | 593 | 1.030 (1.005, 1.054) | 0.017 | |
| Blood urea nitrogen, mg/dL | | | | | 0.502 |
| | Low | 3,337 | 1.051 (1.028, 1.075) | <0.001 | |
| | Medium | 3,366 | 1.044 (1.019, 1.070) | 0.001 | |
| | High | 2,985 | 1.071 (1.053, 1.088) | <0.001 | |
| Albumin, g/dL | | | | | 0.067 |
| | Low | 2,483 | 1.047 (1.032, 1.062) | <0.001 | |
| | Medium | 2,559 | 1.074 (1.044, 1.105) | <0.001 | |

*(Continued)*

**Table 5.** (Continued)

| Stratification Variable | Stratum | n | OR (95% CI) | P Value | P for interaction |
|---|---|---|---|---|---|
| | High | 1,850 | 1.076 (1.030, 1.124) | 0.001 | |
| Hemoglobin, g/dL | | | | | 0.035 |
| | Low | 3,331 | 1.057 (1.042, 1.072) | <0.001 | |
| | Medium | 3,267 | 1.087 (1.062, 1.114) | <0.001 | |
| | High | 3,138 | 1.048 (1.014, 1.084) | 0.006 | |
| Creatinine, mg/dL | | | | | 0.264 |
| | Low | 3,269 | 1.051 (1.027, 1.075) | <0.001 | |
| | Medium | 3,220 | 1.054 (1.029, 1.079) | <0.001 | |
| | High | 3,202 | 1.069 (1.052, 1.087) | <0.001 | |
| Anticoagulant therapy | | | | | 0.185 |
| | No | 9,548 | 1.066 (1.054, 1.079) | <0.001 | |
| | Yes | 188 | 1.123 (1.036, 1.217) | 0.005 | |
| Antiplatelet therapy | | | | | 0.921 |
| | No | 9,137 | 1.068 (1.055, 1.080) | <0.001 | |
| | Yes | 599 | 1.066 (1.021, 1.112) | 0.003 | |

Abbreviations: OR, odds ratio; CI, confidence interval; GCS, Glasgow Coma Scale; APACHE, Acute Physiology and Chronic Health Evaluation.

P interaction values indicate whether the effect modification by stratification variable is statistically significant.

Statistical significance was set at $P < 0.05$ (two-sided).

mediation analysis quantified APACHE-IV's partial intermediary function in the RPR-mortality pathway (proportion: 20.15%), furnishing rigorous statistical scaffolding for pathophysiological inference. This delineates the indirect causal sequence "RPR elevation → exacerbated disease severity → mortality," transcending conventional regression confines to enrich systemic inflammation-prognosis insights.

Notwithstanding compelling results, interpretive caution is warranted due to inherent limitations. Primarily, the retrospective observational design precludes causal inference regarding RPR and in-hospital mortality, with reverse causation (e.g., disease progression precipitating RDW escalation and platelet depletion) unexcluded [5]. Moreover, due to the 2014–2015 eICU data used, the data may not fully capture contemporary management of stroke in the ICU, potentially limiting generalizability to current clinical practice. Secondly, eICU's omission of stroke subtypes (ischemic vs. hemorrhagic), lesion loci, NIHSS scores, and imaging precludes granular neurological impairment adjustment. Third, reliance on admission-singleton RPR measurements neglects temporal fluctuations; Ye et al. [9] and Li et al. [10] affirmed that longitudinal platelet or RPR trajectories surpass baselines in sepsis prognostication, advocating serial assessments henceforth. Additionally, Tong et al. [11] highlighted RPR's independent predictivity in critically ill acute myocardial infarction yet cautioned comorbidity heterogeneity's bias amplification. Our U.S.-centric data may curtail generalizability amid racial and socioeconomic variances, particularly in non-US cohorts. Lastyly, the paucity of RPR-focused systematic reviews or meta-analyses necessitates prospective multicenter corroboration. In addition, although the results are robust, the study's use of a single RPR measurement at admission remains a significant limitation. Future work should obtain multi-timepoint RPR data during hospitalization and apply time-varying analyses to evaluate the incremental prognostic value of RPR dynamics.

RPR is independently associated with in-hospital mortality in critically ill stroke patients, harboring substantial translational promise. Derived from routine complete blood counts, it amalgamates inflammatory (RDW) and hemostatic (platelet) cues, surpassing singular biomarkers in biological coherence and prognostic acuity. Its economical, accessible profile is ideally suited for resource-constrained milieus, facilitating precocious risk stratification, high-acuity triage, and stewardship of surveillance assets—such as vigilant vital-sign oversight, interdisciplinary orchestration, or advancement to escalated

care tiers. Mediation scrutiny posits APACHE-IV as a partial mediator, framing RPR as a holistic disease burden surrogate beyond primordial aberrations. Prospective inquiries might prioritize RPR-steered precision modalities, like subgroup-specific anti-inflammatory or antiplatelet regimens to ameliorate outcomes [8,12,13], alongside multimodal risk-model assimilation to refine stroke-ICU stewardship.

## Supporting information

**S1 Table. Missing data analysis.**
(DOCX)

**S2 Table. Convergence diagnostics of the missForest imputation algorithm.**
(DOCX)

**S3 Table. Comparison of original and imputed data quality assessment.**
(DOCX)

**S4 Table. Results from multiple imputation analysis.**
(DOCX)

## Author contributions

**Conceptualization:** Minmin Lan, Xinghua Qin, Dongmei Yi, Lu Chen.

**Data curation:** Yu Chen, Dongmei Yi.

**Formal analysis:** Xinghua Qin, Lu Chen.

**Methodology:** Xiangrong Yang.

**Project administration:** Dongmei Yi.

**Resources:** Minmin Lan.

**Software:** Minmin Lan.

**Validation:** Xinghua Qin.

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
