## [Decision Letter · Decision Letter 0]

4 Jan 2026

Dear Dr.  Chen,

We look forward to receiving your revised manuscript.

Kind regards,

Marwan Salih Al-Nimer, MD, PhD

Academic Editor

PLOS One

Journal Requirements:

6. Please include captions for your Supporting Information files at the end of your manuscript, and update any in-text citations to match accordingly. Please see our Supporting Information guidelines for more information: http://journals.plos.org/plosone/s/supporting-information....

Additional Editor Comments:

**Major revision**

Reviewer's Responses to Questions

**Comments to the Author**

1. Is the manuscript technically sound, and do the data support the conclusions?

Reviewer #1: Yes

Reviewer #2: Partly

2. Has the statistical analysis been performed appropriately and rigorously?

Reviewer #1: Yes

Reviewer #2: No

3. Have the authors made all data underlying the findings in their manuscript fully available?

Reviewer #1: Yes

Reviewer #2: Yes

4. Is the manuscript presented in an intelligible fashion and written in standard English?

Reviewer #1: Yes

Reviewer #2: Yes

Reviewer #1: This manuscript investigates the relationship between the Red Cell Distribution Width-to-Platelet Ratio (RPR) and in-hospital mortality in critically ill stroke patients using the eICU database. The study addresses a clinically relevant question regarding accessible biomarkers. However, the article has some methodological issues. Specifically, the rationale for covariate selection in the regression model is not sufficiently justified, and the distinction between "association" and "prediction" is blurred. Furthermore, crucial information regarding the calculation of the primary exposure (RPR) and the visual data (Figure 1) is missing or unclear. Given the retrospective nature of the study and the lack of external validation, the authors should refrain from overemphasizing "prediction" and focus on "independent association."

Reviewer #2: The study has the following shortcomings:

This study adopts a retrospective cohort design, which could reveal the association between the red cell distribution width-to-platelet ratio (RPR) and in-hospital mortality in critically ill stroke patients, but cannot establish a causal relationship between them. There is a possibility of reverse causation: the progression of the patient's condition may lead to an increase in red cell distribution width (RDW) and a decrease in platelets, thereby elevating RPR, rather than elevated RPR directly increasing the risk of death. This potential logical relationship has not been effectively excluded.

The study uses data from the 2014–2015 US multicenter eICU database, which is nearly a decade old. The research conclusions drawn based on outdated data have limited applicability in current clinical practice.

The database lacks key neurological information such as stroke subtype (ischemic or hemorrhagic), lesion location, and National Institutes of Health Stroke Scale (NIHSS) score. This makes it impossible to conduct stratified analysis for different subtypes of stroke patients and difficult to fully adjust for the impact of the degree of neurological impairment on prognosis, which may lead to confounding bias in the study results.

The data are only derived from the US population, lacking samples from other ethnic groups (such as Asians, Africans, etc.) and regions. Thus, the applicability of the findings to non-US populations remains unclear.

The study only uses a single RPR measurement within 24 hours of admission as the exposure variable, without considering the dynamic changes of RPR during hospitalization. Relying solely on a single measurement may fail to fully reflect the patient's disease progression and real prognostic risk. The NIHSS score is an important indicator for evaluating the degree of neurological impairment and prognosis in stroke patients, but it was not included as a confounding factor in the adjustment, which may lead to overestimation or underestimation of the association between RPR and mortality.

Ischemic stroke and hemorrhagic stroke differ significantly in pathophysiological mechanisms, treatment strategies, and prognosis, and the predictive value of RPR for mortality may vary between the two. However, the study did not perform subgroup analysis based on this key stratification factor, resulting in insufficient precision of the research conclusions.

The study mentions using the missForest algorithm for multiple imputation to handle missing data, but it does not detail the specific parameter settings of the algorithm, the convergence test results during the imputation process, nor fully compare the differences in data distribution characteristics before and after imputation (only stating that there were no significant differences in key variables). This makes it difficult to evaluate the rationality and reliability of the imputation method.

The study uses a multivariable logistic regression model to analyze the association between RPR and in-hospital mortality, but only reports the odds ratio (OR) and confidence interval without assessing the model's calibration (e.g., Hosmer-Lemeshow test). Thus, the consistency between the model's predicted results and actual observed results cannot be determined.

The study only briefly mentions that RPR integrates pathophysiological processes related to inflammation and coagulation function, but does not deeply analyze the molecular mechanisms and pathophysiological pathways through which RPR affects patient prognosis in combination with the specific pathological mechanisms of stroke, resulting in insufficient theoretical depth of the research.

.

Reviewer #1: No

Reviewer #2: No

---

## [Author Response · Author response to Decision Letter 1]

18 Feb 2026

Dear editors,

Thank you for your letter dated Jan 04 2026 concerning our manuscript titled "The Association between RDW-to-Platelet Ratio and In-Hospital Mortality in Critically Ill Stroke Patients: A Retrospective Cohort Study Based on the eICU Database" (Manuscript ID: PONE-D-25-61250). We appreciate the opportunity to revise and resubmit our work. We are grateful to you and the reviewers for the insightful comments and constructive criticism, which have significantly helped us improve the quality and clarity of our manuscript.

We have carefully considered all the points raised by the reviewers and have revised the manuscript accordingly. Our point-by-point responses to each comment are detailed below, with changes highlighted in the revised manuscript.

Yu Chen

---

## [Editor Report · Decision Letter 1]

20 Feb 2026

The Association between RDW-to-Platelet Ratio and In-Hospital Mortality in Critically Ill Stroke Patients: A Retrospective Cohort Study Based on the eICU Database

PONE-D-25-61250R1

Dear Dr. Yu Chen,

We’re pleased to inform you that your manuscript has been judged scientifically suitable for publication and will be formally accepted for publication once it meets all outstanding technical requirements.

Kind regards,

Marwan Salih Al-Nimer, MD, PhD

Academic Editor

PLOS One
---

## [Editor Report · Acceptance letter]

PONE-D-25-61250R1

PLOS One

Dear Dr. Chen,

I'm pleased to inform you that your manuscript has been deemed suitable for publication in PLOS One. Congratulations! Your manuscript is now being handed over to our production team.

Kind regards,

on behalf of

Professor Marwan Salih Al-Nimer

Academic Editor

PLOS One